# Hypoxia tolerance, but not low pH tolerance, is associated with a latitudinal cline across populations of *Tigriopus californicus*

**Aimee Deconinck** *, Christopher S. Willett

Biology Department, University of North Carolina at Chapel Hill, Chapel Hill, NC, United States of America

* aimeed@live.unc.edu

**Data Availability Statement:** Data has been uploaded to Dryad with the following doi:10.5061/dryad.d7wm37q4p.

## Abstract

Intertidal organisms must tolerate daily fluctuations in environmental parameters, and repeated exposure to co-occurring conditions may result in tolerance to multiple stressors correlating. The intertidal copepod *Tigriopus californicus* experiences diurnal variation in dissolved oxygen levels and pH as the opposing processes of photosynthesis and cellular respiration lead to coordinated highs during the day and lows at night. While environmental parameters with overlapping spatial gradients frequently result in correlated traits, less attention has been given to exploring temporally correlated stressors. We investigated whether hypoxia tolerance correlates with low pH tolerance by separately testing the hypoxia and low pH stress tolerance separately of 6 genetically differentiated populations of *T. californicus*. We independently checked for similarities in tolerance for each of the two stressors by latitude, sex, size, and time since collection as predictors. We found that although hypoxia tolerance correlated with latitude, low pH tolerance did not, and no predictor was significant for both stressors. We concluded that temporally coordinated exposure to low pH and low oxygen did not result in populations developing equivalent tolerance for both. Although climate change alters several environmental variables simultaneously, organisms' abilities to tolerate these changes may not be similarly coupled.

## Introduction

Physiological tolerance is essential in highly variable environments, and few marine inhabitants experience as much routine environmental change as residents of the rocky intertidal habitat. High intertidal rock pools experience daily and seasonal changes in temperature, dissolved oxygen (DO), and pH rarely seen in other marine systems [1–3]. In surface aquatic systems, DO and pH covary temporally as the opposing processes of photosynthesis and aerobic respiration drive the exchange of oxygen and carbon dioxide gases. During the day, photosynthetic organisms use dissolved carbon dioxide gas for photosynthesis and release oxygen gas, raising DO levels. The consumption of dissolved carbon dioxide also shifts the equilibrium of the $CO_2$-bicarbonate-carbonate system, increasing the pH, although the shift is slower than the oxygenation rate due to the different diffusion rates of the two gases [4, 5]. At night,

**Funding:** This research was supported in part by two grants from the National Science Foundation [IOS-2029156 and IOS-1555959 to J.G. Kingsolver and C.S.W.]. https://www.nsf.gov/ The funders had no role in study design, data collection and analysis, decision to publish, or preparation of the manuscript.

**Competing interests:** The authors have declared that no competing interests exist.

respiration consumes oxygen and releases carbon dioxide, reversing the pattern to the point where waters may become hypoxic (DO≤2.0 mg $O_2$/L or 30% saturation) and slightly acidic (pH<7) [2]. Thus, given the typical correlation in environmental conditions, we might expect tolerance to decreased oxygen levels and decreased pH to correlate as well [6–8].

Species that are distributed widely across a geographic gradient frequently develop locally adapted populations [9–12]. *Tigriopus californicus* [13] inhabit high intertidal splash pools from the central Baja Peninsula in Mexico to southern Alaska in the US, and over this range they experience a wide range of physiochemical conditions. Maximum daily temperatures can exceed 40˚C at the southern limit of its range [14], and minimum daily temperatures can descend below 10˚C at the northern limit [15], Similarly, salinity may increase to 100 ppt in the southern regions during periods of evaporation [14] or dip to 11.5 ppt during periods of precipitation in the north [15, 16]. These geographic differences in habitat combined with low dispersal have resulted in populations with highly divergent genomes, particularly for mito-chondrial DNA [17–19] (S1 Table). Common garden experiments have demonstrated local adaptation to thermal tolerance and some measures of salinity tolerance align with latitude, as southern populations are more tolerant of heat shock [20–23] and northern populations, which receive more rain, are generally more tolerant of extended periods at low salinity [14, 15, 24].

Although we are unaware of any studies specifically exploring latitudinal clines of hypoxia and low pH adaptation, multiple studies have investigated the correlation between tempera-ture clines and hypoxia or low pH tolerance [25–29]. Warmer water dissolves less oxygen than colder water which implies that elevated temperatures can increase hypoxia and acidic condi-tions within tidepools resulting in more stressful conditions where water is warmer. Since studies exploring hypoxia and low pH tolerance with latitude were absent, we used thermal tol-erance studies to inform our hypotheses. For example, *Tigriopus brevicornis* from Scotland consumed less oxygen at low temperatures while experiencing hypoxia, but at the highest tem-perature the level of oxygen needed to maintain oxygen regulation ($P_{crit}$) was significantly ele-vated [30]. Although *T. californicus* from Washington survived high temperature stress approximately the same regardless of DO levels, the lowest DO level tested was 22% of satura-tion for 5 hours [31]. In another study, *T. californicus* copepods survived at least 24 hours with no oxygen [32], which may indicate that the DO levels tested by [31] were not stressful. Several copepod species exhibit signs of sublethal stress to ocean acidification when it is paired with higher temperatures [33, 34] although *T. californicus* may be adapted to lower pH than is com-mon in coastal waters [35]. Thus, we expected that southern populations of *T. californicus* would experience more intense hypoxia and low pH stress and would have higher tolerance to the stressors than northern populations.

In addition to latitudinal clines in environmental parameters, stress tolerance may also be associated with sex-specific adaptation, laboratory adaptation and changes in life history traits. Sex-specific variation in environmental stress tolerance is well documented, and, in *T. califor-nicus*, males are often the more sensitive sex [36], yet models comparing directional to cyclic environmental change predict that the difference between sexes should decrease with rapidly changing environments [37]. Adaptation to laboratory conditions is also well documented in multiple systems, as populations respond to different selective pressures than they experienced in the field [38–40]. In contrast to the unidirectional relationship between stress and sex-spe-cific or laboratory adaptation, body size differences may be both a response to environmental stress [41–43] as well as a predictor of thermal tolerance [44, 45], although its impact on other forms of stress are less clear [46–48]. Since multiple predictors may influence stress tolerance, this paper aims to determine which variables are most important for predicting hypoxia and low pH tolerance in genetically divergent populations of *T. californicus*.

We tested adult copepods for either hypoxia tolerance or low pH tolerance to determine if the diurnal correlation of the stressors resulted in phenotype correlation. Here we define correlated evolution as two or more traits evolving synchronously with each other [49–52]; specifically, this definition notes the observed pattern without assumptions of a mechanistic link between the traits being measured. We also distinguish it from coadaptation which implies a direct link between two evolving traits and note that correlated evolution would be a necessary pattern for coadaptation to occur. We expected (1) the co-occurrence of these stressors temporally would result in population-specific correlated tolerance to each in single-stressor studies, (2) hypoxia tolerance and low pH tolerance separately would inversely correlate with latitude, and (3) sex, body size, and time since collection would have significant effects on the tolerance for each stressor independently.

## Methods and materials

### Field measurements

Sunrise and sunset measurements of temperature and pH were taken for five days with an Oakton Waterproof pH Testr® 30 for 6 rock pools that contained copepods at Bodega Head State Beach (38.30466, -123.06532) from Dec. 30, 2017 to Jan. 3, 2018 to provide an estimate of the spatial and diurnal variability in pH experienced by *T. californicus*. These pools were along the exposed shoreline at Bodega Head whereas collections were from Bodega Marine Lab.

A follow-up study took temperature, salinity, DO and pH measurements at four locations along the California coastline during August 2021: Bodega Head State Beach (38.30466, -123.06532), Santa Cruz (36.949783, 122.047033), Abalone Cove Shoreline Park (33.737456, 118.373652), and Ocean Beach (32.745682, 117.255103) (Fig 1 and S2 Table). While we recognized our study was not comprehensive enough to depict annual variability at any single location, we expected that any large latitudinal trends in DO and pH might be captured by sampling across locations at similar times. This was an ambitious goal, but we could find no data on latitudinal differences in DO and pH for rock pool habitats. The data presented herein are not meant to be ultimate descriptions of rock pool conditions (e.g. we used hand probes instead of titration procedures for estimating DO and total alkalinity in combination with pH); rather, they are meant to be exploratory samples to provide direction for follow-up studies. Measurements were taken with a Fisherbrand™ Traceable™ Portable Dissolved Oxygen Meter Pen (0–20 mg/L range and ±0.4 mg/L accuracy) and the previously mentioned pH meter at 5 rock pools that contained copepods for 3 to 4 days at each location. Both instruments were calibrated daily following the manufacturers' recommendations. At least two time-points (morning and afternoon or evening) per day were measured as health and safety permitted. San Diego was the only location at which three measurements per day could be taken. Data for pH at Bodega and Santa Cruz were limited due to an instrument malfunction and delays in shipping for a replacement.

### Collections

Populations of *Tigriopus californicus* were collected between September 2004 and April 2018 from 6 rocky intertidal locations along the eastern Pacific coastline: Friday Harbor Laboratories, WA (FHL); Bodega Marine Lab, CA (BB); Santa Cruz, CA (SCN); San Simeon, CA (SS); Abalone Cove, CA (AB); and San Diego, CA (SD) (Fig 1 and S2 Table). After maintaining populations under the same environmental conditions for at least 6 generations, Individual copepods were used only once in a single-stressor trial for the hypoxia assay, low pH assay or control run. This study was intended to identify patterns of local adaptation rather than the interactive effects of multiple stressors, so only single-stressor trials were conducted.

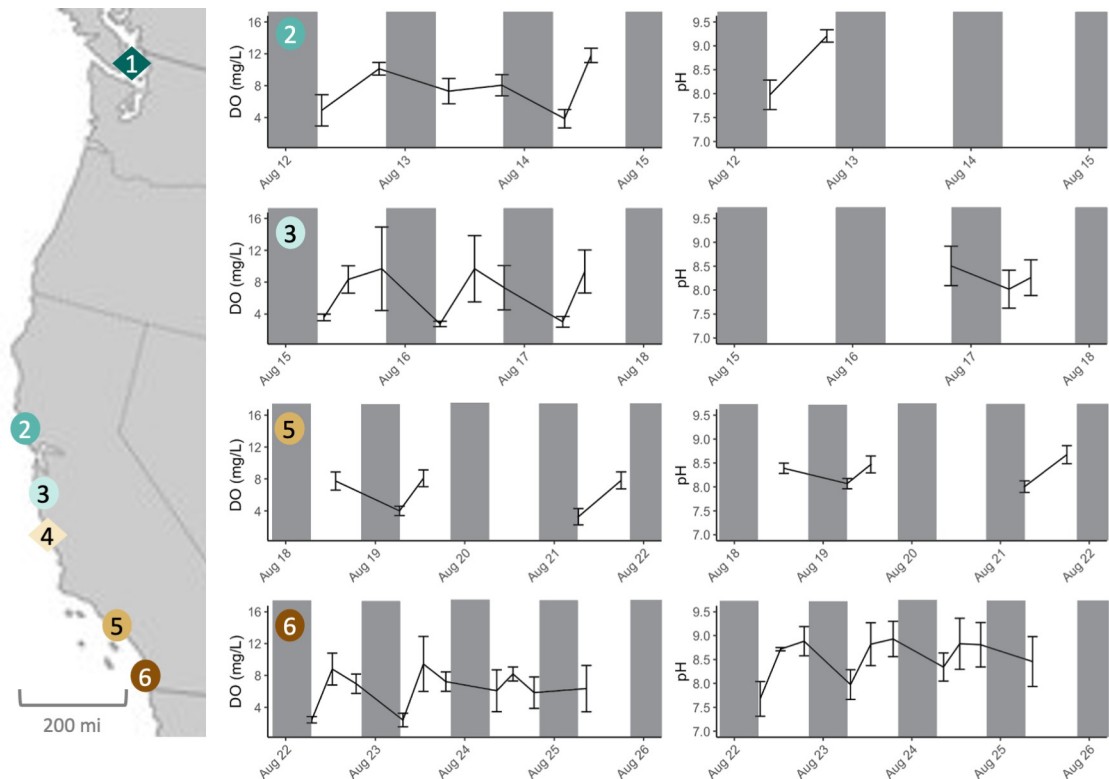

**Fig 1. Map of collection locations and mean tide pool DO and pH during August 2021.** Copepods from six locations along the Pacific West coastline (1 = Friday Harbor Laboratories, WA; 2 = Bodega Marine Lab, CA; 3 = Santa Cruz, CA; 4 = San Simeon, CA; 5 = Abalone Cove, CA; 6 = San Diego, CA) were collected and transported to the University of North Carolina at Chapel Hill. Of these, four (indicated by circle shapes) were visited during August 2021 to estimate DO and pH variability. Five rock pools at each location were sampled for 3–5 days in the morning and afternoon or evening. Means are plotted with s.d. bars. The left column of graphs shows DO measurements, and the circled number in the top left corner of each plot indicates the corresponding location on the map to the left. The right column of graphs shows pH measurements. An instrument malfunction resulted in missed sampling for pH for four days in the top two pH plots.

## Copepod rearing

Once at the University of North Carolina at Chapel Hill, collections were transferred to 400 mL beakers and fed ground commercial fish flakes and natural algae growth *ad libitum*. Salinity was maintained at 35 PPT using artificial seawater, and cultures were kept in incubators at 20°C with a 12-h light/dark cycle. Collection ages ranged from 1 to 15 years at time of testing (S2 Table).

For hypoxia and low pH assays, gravid females were removed from mass culture beakers and placed individually in 24-well plates or in groups of 10 in petri dishes and fed ground fish flakes. When nauplii were visible, the female was removed, and the nauplii were allowed to mature.

## Glovebox construction

Arenas for holding up to five copepods per trial were made from a plastic cylinder (3-inch diameter; 4-inch height) with nylon mesh (48 micron) glued to the bottom (S1 Fig). A 3-L plastic water bath was filled with seawater and could hold 8 arenas at a time. Temperature in the bath was maintained by a loop filled with water cooled by a cooling tower. Due to the possible exchange of gases with the atmosphere, we did not allow the water in the water bath to be

directly circulated through the cooling tower. Gases were delivered via an airstone placed at the bottom of the water bath, and a magnetic spin bar at the bottom of the water bath circulated the water among all of the arenas.

The glovebox was constructed using ¼" acrylic and laser cut following the blueprints which have been deposited on Dryad and sealed with clear silicone caulk. Four spring hasps were attached for the door using screws and bolts, again sealed with silicone caulk. The door was lined with self-adhesive weather stripping to insulate the door-box juncture when in use. Two 4" x 3.52" diameter coupling PVC fittings, sealed with silicone caulk, were used to secure two long kitchen gloves with hose clamps.

### Hypoxia and low pH assays

Hypoxia tolerance and low pH tolerance were measured in single-stressor assays conducted in the custom-built glove box described above (S1 Fig) at the University of North Carolina at Chapel Hill. During each assay, up to five copepods were grouped by population and sex and placed in a single arena made of a plastic cylinder with a fine mesh bottom. Eight arenas were placed in a water bath that was maintained at 20°C (±2°C) and enclosed by the glove box. Populations and sexes were assigned randomly to one of the eight arenas during each trial to minimize placement effects. In all assays, animals were not fed during the gas adjustment period or the experiment.

For the hypoxia assay, nitrogen gas was bubbled continuously to displace dissolved oxygen and maintain a low oxygen environment followed by a reoxygenation period before measuring survival (S2 Fig). Since this method increases the pH as carbon dioxide is displaced, we conducted exploratory experiments of low oxygen and high pH conditions separately. We bubbled nitrogen continuously at a very low rate and found that *T. californicus* were highly tolerant (survived more than two days) of DO levels greater than 0.1 mg/L (S3 Table). We also used sodium hydroxide to increase the pH and found that *T. californicus* could survive pH higher than 10.1 for more than 24 hours (S3 Table). As a result, the hypoxic period was started when the DO level was ≤0.05 mg/L which took 1.98±0.60 hours of bubbling nitrogen gas; this gas adjustment period was not included in the experimental time. During these studies, pH was not measured, but later replicates using the same setup and conditions, which included pH measurements, found that pH increased to a maximum of 9.8. DO ≤0.05 mg/L was maintained by continuous nitrogen bubbling for 20 hours because this was when more than half of the copepods stopped swimming. Dissolved oxygen was sampled every 5 minutes during the experiment using an Orion Star A213 Dissolved Oxygen (DO) Benchtop Meter (Thermo Scientific) which can measure 0–50 mg/L with ±0.1 accuracy. So long as the stream of nitrogen gas was maintained, the DO level remained ≤0.05 mg/L, and only the measurements at the start of the hypoxic period, end of the hypoxic period, and end of the reoxygenation period were saved. The DO probe was cleaned monthly and calibrated weekly with air and yearly with a zero-oxygen solution as recommended by the manufacturer. After the hypoxic period, atmospheric air was pumped into the box for 10 hours using an aquarium air pump, restoring the DO level to 7.90±0.46 mg/L within 10 minutes, then the number of surviving individuals were counted by gently prodding individuals with a pipette tip to elicit an evasion response (swimming away).

For the low pH assay, carbon dioxide was bubbled into the water bath until the pH measured 4.8 then the water bath was allowed to equilibrate with the atmosphere for 24 hours before measuring survival (S2 Fig). Preliminary work showed that although pH 5.5 knocked down <10% of copepods, 1 h at pH 5.0 knocked down >90% (S3 Table). Additionally, measurements taken at Shore Acres and Cape Arago State Parks in Oregon during spring months

average pH 6.0 with a minima of 5.0, thus our choice of 4.8 as a target pH, although extreme for organisms in coastal waters may not be so extreme for *T. californicus* [53]. The final time point was set as 24 h because there was little change from the number of recovered copepods after 12 hours. When the pH reached 4.8 which took less than 10 minutes, the gas supply was turned off, and the water bath was allowed to return to normal pH over time (S3 Fig). A Fisherbrand Accumet AB15 Basic and BioBasic pH/mV/˚C Meter was used to measure pH, and the previously mentioned DO meter was used to measure DO. The pH meter was calibrated weekly using 3-point standard pH buffer solutions following manufacturer guidelines. The number of swimming individuals, temperature, pH and DO were sampled every 20 minutes for the first 6 hours, then again at 12 hours and 24 hours. Dissolved oxygen levels did not drop below 1.85 mg/L during any low pH trial (S3 Fig). Although this DO level is hypoxic, oxygen levels rose quickly, and only half of the trials experienced low pH and low DO together at all. Given the short length of time (less than 1 hour) and that half of the copepods in the pH assays never experienced the two stressors together, we treated all trials as single-stressor pH assays. As with the hypoxia assay, survival was assessed by gently prodding individuals with a pipette tip to elicit an evasion response (swimming away).

Control trials for both assays were set up with the same materials and conducted at the same time as the hypoxia and low pH assays except atmospheric air was added using an aquarium air pump rather than nitrogen or carbon dioxide gas. Due to the limitation of equipment, DO and pH for control trials were only sampled at the beginning and end of an assay although values remained consistent (pH = 7.81±0.30; DO = 8.22±0.26 mg/L). When possible, siblings from the same brood were split between control and experimental assays. Although survival was the only fitness variable we measured for all assays, survival is essential for reproduction and other measures of fitness. Furthermore, as no other study had explored whether populations locally adapted for hypoxia tolerance are also adapted for low pH tolerance, we chose to focus on identifying other covariates that could explain the correlation rather than exploring the effects of the correlation on other life history traits.

After each assay, individual copepods were photographed using either a handheld digital microscope (Amscope) or a digital camera attached to a microscope (OMAX), and their body length was measured from the cephalasome to the caudal ramus using the segmented line tool in Fiji [54]. Since copepods were not tracked individually during assays, only the survival at the final time measurement could be used in generalized linear mixed models in order to include the length measurement for each individual as a fixed effect. Any copepods that could not be recovered at the end of the assay because they became stuck in tiny bubble pockets of the epoxy glue were excluded from final analysis. In total, this accounted for 2.4% of our sample across both experiments.

## Statistical analyses

Descriptive statistics (mean and s.d.) for each location were calculated across the sampled tide pools and aggregated by sampling time (morning, afternoon, evening). Although the absolute lowest DO and maximum pH levels likely occurred in the 30 minutes prior to sunrise, we did not sample before sunrise out of concern for the safety of the observer who had to physically hike and climb to reach the sampling locations for each observation time point. We then generated a linear model each for DO and pH with latitude-by-sampling-time as the only predictor. Next, we ran ANOVAs on each of the fitted models to test for latitudinal trends for morning, afternoon, and evening measurements of DO and pH.

An ANOVA was also conducted for the hypoxia and low pH assays to verify differential survival between treatment and control groups.

Linear regression was used to measure the regression coefficient of each predictor in R version 3.6.2. Tolerance to hypoxia and low pH were measured as survival after treatment and analyzed using a generalized linear mixed effect model with a logit link for binomial distribution from the lme4 package [55]. Body length was log transformed, and all continuous variables (latitude, collection year, and log-length) were centered and scaled prior to analysis by dividing each value by twice the standard deviation [56]. Collinearity amongst predictors was checked using the car package [57] and predictors with a variance inflation factor>5 were removed from the full model. The full model for both hypoxia and low pH tolerance included latitude, collection year, sex and log-length as fixed effects, and the random effects included position and group (individuals in an arena) nested within batch (arenas in the same assay). The greater sample size in the hypoxia tolerance studies provided sufficient power to allow the inclusion of all two-way interactions. After verifying and correcting for the model assumptions, the full models were dredged using the MuMIn package [58]. The model with the lowest AICc was selected as the estimated best fit model, and models that had some support (ΔAICc<5) were averaged with the estimated best fit model. Coefficient values, confidence intervals and p-values were computed from the averaged model using Satterthwaite's method and t-statistics. To further examine the support of each predictor in the final averaged model, the weighted importance (I) of each variable—the sum of model weights across all models that included the variable—was calculated [59].

To test for correlation between hypoxia tolerance and low pH tolerance, a Pearson correlation test was performed on the population-level response. Mean survival of each population was used because hypoxia and low pH tolerances were not tested on the same individuals.

### Ethics statement

The organisms used in this study were exempt from approval by the Institutional Animal Care and Use Committee.

## Results

Measurements of pH taken at Bodega Head State Beach during the winter of 2017/2018 revealed that pH was lower at sunrise (7.60±0.03) than at sunset (8.53±0.03) and differed from sunset measures of coastal water pH (7.75±0.02). Overall, pH changed an average of 0.93 units per day with a maximal change of 1.32 units (S4 Fig). Diurnal changes in pH were large at nearly a 10-fold change from sunrise to sunset, much greater than what is typically experienced by subtidal marine organisms only meters away [60]. Furthermore, these conditions frequently differed from the nearby water column, such that the timing of tidal flushing could rapidly (within minutes) alter the conditions within pools. Measurements of pH during August 2021 again showed that pH was lower in the morning and higher in the evening, and daily changes in pH at San Diego, the most southern location, were comparable to those observed at Bodega Head State Beach, the most northern location sampled (Fig 1 and S4 Table). We detected a small but significant difference in morning pH values for each location ($p = 0.028$; β = -0.04376; 95% CI±0.0389), but we did not detect significant differences in pH for afternoon ($p = 0.148$; β = -0.02907; 95% CI±0.03959) or evening ($p = 0.269$; β = -0.02180; 95% CI ±0.03897) timepoints. Generally, as latitude decreased, pH increased with the most substantial difference occurring in the morning.

DO measurements during August 2021 revealed that oxygen levels were lower in the morning (4.13±2.03 mg/L), reaching a peak in the afternoon (9.04±2.39 mg/L) before declining slightly in the evening (7.88±2.54 mg/L). At some locations, DO could vary by up to 48% across pools measured at the same location and same time. Morning ($p = 0.018696$; β =

0.19068; 95% CI±0.15844), afternoon ($p<0.001$; β = 0.33388; 95% CI±0.16067), and evening ($p<0.001$; β = 0.29816; 95% CI±0.15918) timepoints were all significantly different across latitude for DO. As latitude increased, so did DO levels with the greatest difference across locations occurring in the afternoon. (Fig 1 and S4 Table).

We measured hypoxia tolerance of 696 copepods from 6 genetically differentiated populations of the copepod *T. californicus* by measuring survival after 20 hours of hypoxia exposure and 10 hours of reoxygenation. The hypoxia assay resulted in significantly lower survival ($F_{1, 694}$ = 323.7, $p<0.001$) than the control. We then estimated how the odds of survival changed by averaging the best fit regression models (ΔAICc<5) and weighting the importance of each variable by the frequency of their presence in the best fit models. We found that latitude ($p<0.001$; β = 7.82226; 95% CI±3.0819144; I = 1.00*), length ($p<0.001$; β = -1.435299; 95% CI ±0.7597687; I = 1.00), and year ($p<0.001$; β = -5.127622; 95% CI±2.16483; I = 1.00) as well as a latitude-by year interaction ($p<0.001$; β = -6.259887; 95% CI±2.5606168; I = 1.00) were both significant and important for the averaged model (S5 Table). Sex ($p$ = 0.545342; β = -0.2156621; 95% CI±0.95505; I = 0.63) was also a relatively important predictor although it was not significant in the final averaged model. Other interactive terms, such as sex-by-latitude, were neither significant nor important. Tolerance to hypoxia increased with increasing latitude (Fig 2), and females were more tolerant than males in 5 of the 6 populations (Fig 3). Tolerance to hypoxia decreased with increased length and more recent time since collection, although the latter was opposed by the effect of latitude. In other words, older collections also tended to be southern populations, resulting in a significant latitude-by-year interaction as sensitivity to hypoxia stress was opposite for these two predictors.

Similarly, we measured low pH tolerance of 472 copepods across this same set of populations by reducing the pH to 4.8 and measuring recovery (swim when prodded) after 24 hours.

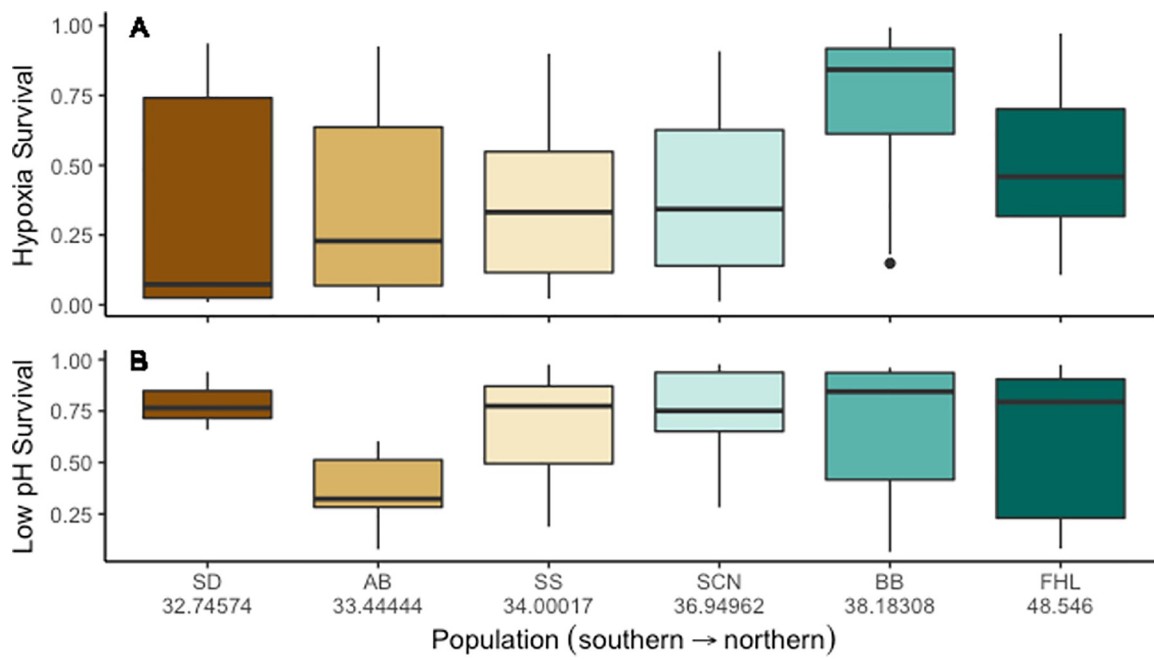

**Fig 2. Probability of survival by population of *T. californicus* for hypoxia and low pH stresses.** Panel (A) is the probability of survival for hypoxia stress. Panel (B) is the probability of survival for low pH stress. Latitudes are written below population abbreviations in (B). Populations are ordered but not scaled by latitude. The medians are presented with the 25th and 75th percentiles as the box and 10th and 90th percentiles as whiskers. Populations performed significantly different across latitude during hypoxia assays but not low pH assays.

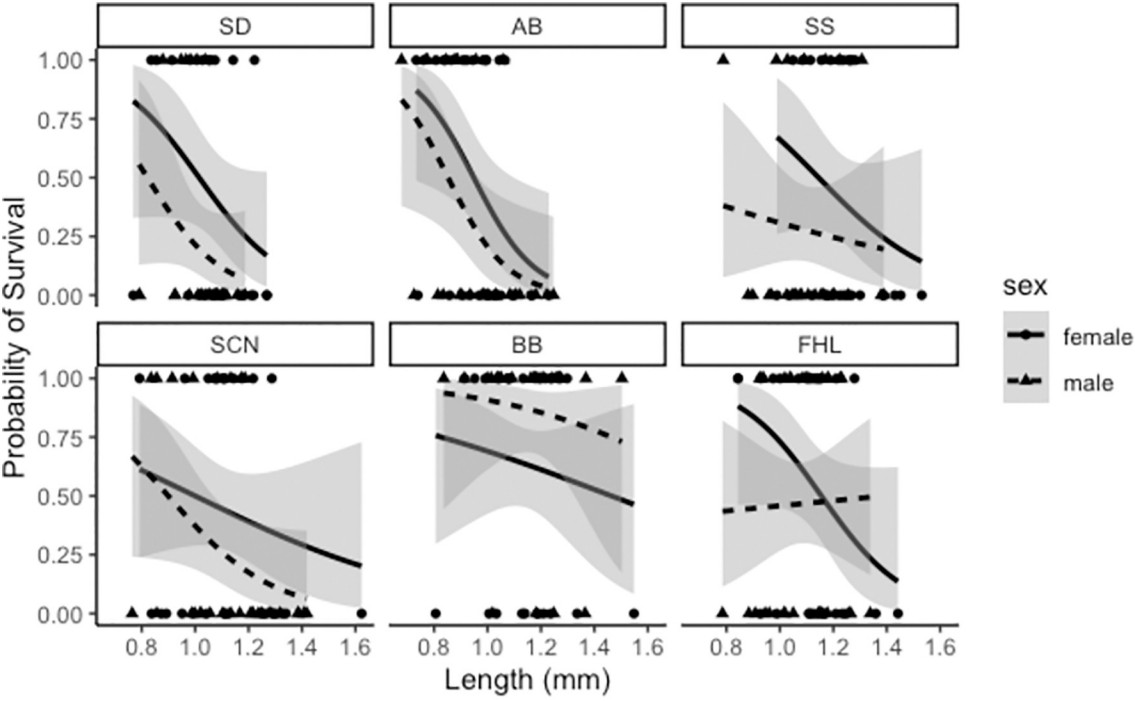

**Fig 3. Effect of length, sex and population organized by collection date on hypoxia tolerance in *T. californicus*.** The lines represent the averaged hypoxia model fit with shaded s.e.m. bars. Points represent actual scores (1 if survived and 0 if dead). Females are solid lines and circles, and males are dashed lines and triangles. Panels are arranged from southern most (SD) to northernmost (FHL). Latitude, length, collection year and latitude × year were significant and important effects. Sex was important but not significant.

The low pH treatment resulted in significantly lower survival ($F_{1, 470}$ = 99.53, $p$<0.001). For low pH tolerance, the only significant predictor was sex ($p$ = 0.0404; β = -1.614681; 95% CI ±1.4749847; I = 0.90), and sex was the only predictor with high importance across all models (S6 Table). Time since collection ($p$ = 0.4828; β = 0.4731268; 95% CI±0.829561; I = 0.49) was marginally important, and the predictors body length ($p$ = 0.9772; β = 0.008333252; 95% CI ±1.132061952; I = 0.26) and latitude ($p$ = 0.9777; β = -0.01100642; 95% CI±1.56668978; I = 0.27) were less important. Males were the more sensitive sex for all but one population (Fig 4), and low pH tolerance increased with more recent collections. The effects for latitude and body length were indistinguishable from zero.

The Pearson correlation test for hypoxia and low pH tolerance of each population suggests no correlation between the traits (t = -0.38822, $p$ = 0.7176; Fig 5). This lack of correlation was also reflected by the absence of overlap of predicting variables in models for the two traits. Whereas latitude, length and collection year exhibited significant linear trends for hypoxia survival as described above, none of these variables significantly predicted the odds of survival for copepods in the low pH assay. Likewise, sex, the only significant variable in the low pH assay, did not significantly predict the odds of survival for copepods in the hypoxia assay.

## Discussion

Our field measurements of pool conditions indicate that photosynthetic and respiration rates, rather than temperature, appear to be the primary drivers of pH and DO variability observed within the high intertidal pools, and these results are consistent with other research measuring low DO and pH covariance [61–67]. Our expectation was that pH and DO would be lower at sunrise and highest at sunset. While pH followed our prediction, DO did not, reaching its

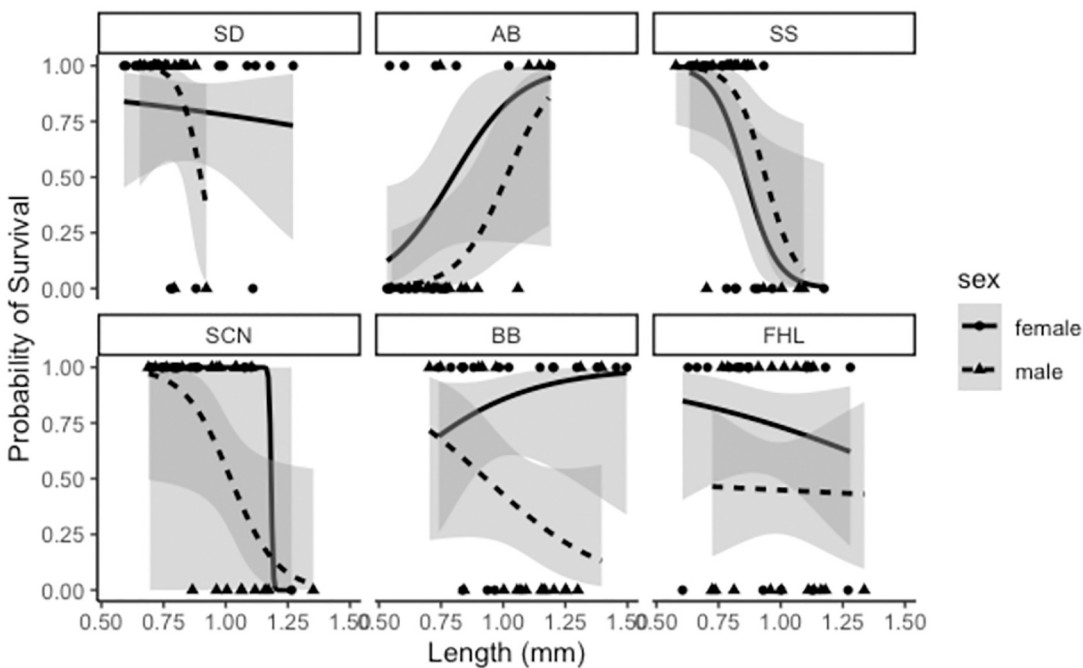

**Fig 4. Effect of length, sex and population organized by collection date on pH tolerance in *T. californicus*.** The lines represent the averaged low pH model fit with shaded s.e.m. bars. Points represent actual scores (1 if survived and 0 if dead). Females are solid lines and circles, and males are dashed lines and triangles. Panels are arranged from southernmost (SD) to northernmost (FHL). Sex was the only significant and important effect.

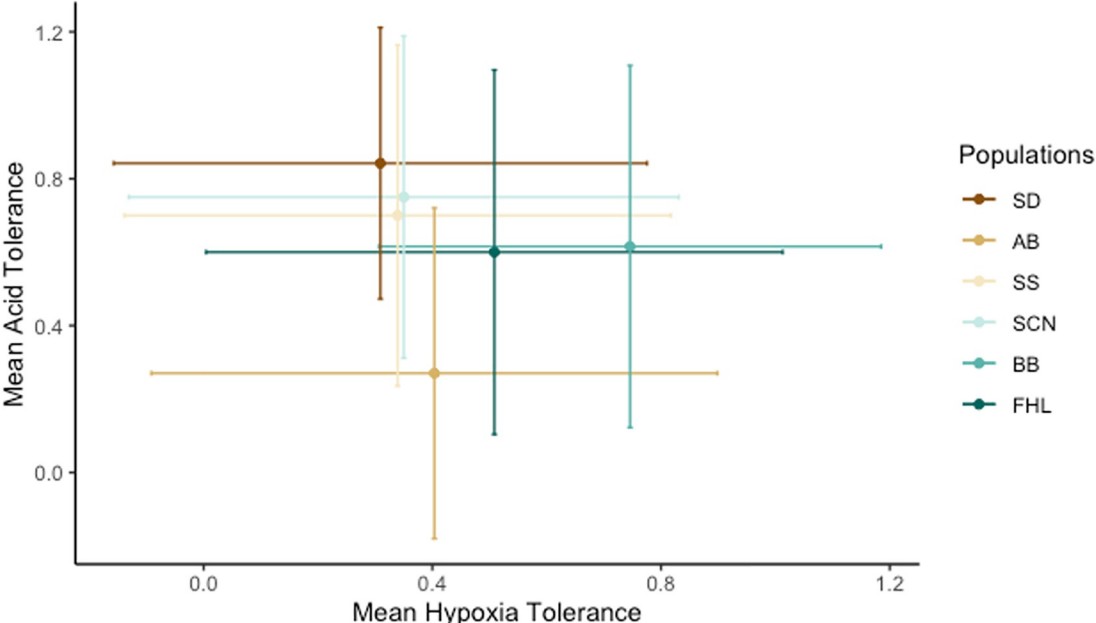

**Fig 5. Correlation of hypoxia and pH tolerance for populations of *T. californicus*.** The responses for hypoxia and low pH tolerance were compared using a Pearson correlation test. Points are the means of the predicted values from the averaged models of hypoxia tolerance and low pH tolerance by population. They are displayed with s.d. bars.

maxima in the afternoon rather than at sunset. The oxygen maxima could be a result of day-light intensity being greatest at midday driving the greatest release of oxygen into the water at that time. On the other hand, photosynthesis does not directly alter pH; instead, the process may lag as dissolved carbon dioxide is restored by reestablishing equilibria between several carbon moieties, overall resulting in the pH maxima at sunset [5, 68]. Our measurements did not include sampling throughout the night, however, and such sampling is necessary to definitively establish when oxygen and pH minima occur.

Our samplings were also not comprehensive enough to record annual patterns in DO and low pH fluctuations. Our morning measurements were rarely hypoxic, but we only measured DO during the summer when daytime is longer. Seasonality and latitude may significantly interact to generate complex trends in DO and low pH. Daylight length is shortest during late fall and winter, and northern latitudes have shorter daylight lengths during the winter than southern latitudes. It would be interesting to sample DO levels during late fall and early winter across locations to see if a latitudinal trend in oxygen minima could be detected then. Again, we reiterate that our measurements for DO and pH did not include nighttime measures, and therefore do not include the lowest possible DO and pH measures. Even so, our data clearly illustrate the differences between in DO and pH of intertidal pools (Fig 1 and S3 Table) and the nearby water column which have only slight changes in dissolved oxygen [69] and pH [70] with latitude. We encourage additional research into the physiochemical conditions of the rock pool habitat as a possible model of extreme aquatic conditions.

After measuring survival for hypoxia and low pH in *T. californicus* copepods, we found no evidence to support our first hypothesis that temporal correlation of two environmental stressors would result in correlation in tolerance for them suggesting that the mechanisms for tolerance to each stressor are similarly distinct. The finding that hypoxia and low pH tolerance is uncorrelated was surprising given how frequently these stressors covary in aquatic habitats[4] (Fig 1). The absence of phenotypic correlation strongly suggests that the molecular mechanisms leading to tolerance in these two traits are unrelated as well. Hypoxia survival in *T. californicus* likely relies on adjusting the permeability of the chitin exoskeleton to oxygen and glycolytic energy production since the species lacks respiratory pigments and the HIF-1 pathway [32]. Additionally, hypoxia stress resulted in the upregulation of numerous mitochondrial proteins for metabolic processes and heat-shock proteins [32]. Perhaps counterintuitively, many hypoxia-sensitive species down regulate metabolic activity of the mitochondria during severe hypoxia stress while hypoxia-tolerant species continue to rely on mitochondrial proteins for metabolic intermediates [71]. The presence of an alternative oxidase gene in *T. californicus* may explain how their mitochondria are able to avoid redox imbalance while operating when oxygen is unavailable [72]. Heat-shock proteins have been implicated in tolerating a number of processes beyond thermal and hypoxic stress including environmental pollutants [73], desiccation [74], osmolarity [75], as well as ammonia, salinity, and pH [76] to name a few. In contrast, exposure to slightly acidic conditions for 24 hours resulted in a different transcriptomic response—the upregulation of oxidative stress proteins—in *Tigriopus japonicus*, a sister species of *T. californicus* [77]. An organism's response to low pH conditions can also overlap with osmoregulation pathways, such as sodium:proton exchange [78] and ion homeostasis [79]. This experiment does not support that temporal covariance results in correlated adaptation as strongly as geographic correlation has for some other traits and species [80–83]. Our study is unique for exploring the correlation of survival to low oxygen and low pH with latitude, but additional research with other life history traits may come to a different conclusion [5, 84].

The population-specific differences in hypoxia survival, but not low pH survival, strongly correlated with latitude, which differed from our second hypothesis that both tolerances would

correlate with latitude. This suggests *T. californicus* exhibit a phenotypic gradient for hypoxia survival, as has been documented for other environmental parameters in this species [14, 20], and any phenotypic cline in low pH survival is substantially weaker. The latitudinal clines in thermal tolerance [14, 20–23] and hypoxia tolerance inversely covary, which did not match our prior expectations. The oxygen-and capacity-limited thermal tolerance hypothesis predicts that an ectotherm's thermal tolerance is limited by its ability to deliver oxygen to meet metabolic demand [25, 27, 85–87] although see also [88–90]. *T. californicus* rely on simple diffusion from the environment to meet their oxygen needs, and we anticipated that the pattern of their thermal tolerance reflected limits of available DO at lower latitudes. In contrast, we observed that the warmest part of the day (afternoon) also had the highest DO levels suggesting that photosynthetic and respiration rates are stronger drivers of available oxygen than temperature during the summer. The apparent inverse temporal correlation across a 24-hour period between oxygen and temperature and, to a lesser extent, low pH and temperature may explain the inverted patterns of latitudinal tolerance (for hypoxia and thermal tolerance) or absence of a pattern (for low pH and thermal tolerance). As mentioned previously, *T. californicus* have also been well studied for divergence in the mitochondrial genome with latitude, and the mitochondrial genome is integral to hypoxia tolerance [91]. Since our current study did not include genomic analysis, it would be interesting for future studies to explore this possible link. Of final note, the latitudinal trend for hypoxia survival is also not consistent with our DO data, which indicates lower morning DO values at lower latitudes, and our pH data is similarly inconsistent with the observed phenotypic pattern in low pH tolerance. We are reluctant to draw a strong inference from these observations as our sampling was not comprehensive, but we mention them here to draw attention to the need for rigorous sampling of latitudinal variability in the rock pools that *T. californicus* inhabit.

Our third and final hypothesis—that sex, body size, and time since collection would also significantly change the odds of survival for hypoxia and low pH—also differed from our results. No other predictor changed the odds of survival for hypoxia tolerance as strongly as latitude, but sex was also an important predictor for hypoxia tolerance and the only significant predictor of low pH tolerance. In both cases, males were more sensitive than females, which is consistent with previous studies in this system [36, 92]. *T. californicus* copepods lack heterogametic sex chromosomes meaning that any sex-specific differences in survival cannot be explained by asymmetric inheritance of sex chromosomes but does not rule out asymmetric inheritance of the mitochondrial genome [93] or potential downstream effects of sex-specific endocrine expression [94].

Larger individuals were more sensitive to hypoxia, but length was not a significant predictor for low pH survival. As mentioned previously, *T. californicus* rely on cuticular diffusion for gas exchange [32], so larger body size would reduce the surface area available for gas exchange relative to body volume. Additionally, more mass would require more oxygen to sustain metabolic functions which could explain the increased sensitivity to low oxygen [95]. Given that females and northern populations were both larger yet more hypoxia tolerant, size does not appear to affect hypoxia survival as strongly as other predictors which is consistent with its smaller regression coefficient.

Time since collection was only significant for models of hypoxia survival. Older collections were less sensitive to hypoxia which suggests that the conditions normally employed for maintaining *T. californicus* in the lab may inadvertently be selecting for hypoxia tolerance. Each population, however, was collected only once, confounding our interpretation of this effect with latitude. There was a significant and negative interaction between latitude and time since collection, because northern populations, which tended to be more tolerant, were also newer collections, which tended to be less tolerant. Consequently, the observed trends in hypoxia

tolerance may differ from those represented in this data set, but a clearer interpretation would require repeated tests on the same population with different collection times.

*T. californicus* copepods experience a highly variable environment. Prior studies of *Tigriopus* habitats found similar or greater changes in pH than coastal waters with correlated decreases in DO [1, 2, 16]. This study utilized environmental conditions outside of ranges typically experienced in native *T. californicus* habitats in order to generate observable responses rather than determining physiological limits. It is not uncommon to expose organisms to environmental conditions beyond their normal experience when assessing the limits of their tolerance [96, 97]. Our experiments also did not explore other fitness consequences of hypoxia and low pH stress, Individuals that experience sublethal stress may exhibit reduced fecundity [15, 77, 92, 98, 99], increased development time [15, 77, 99, 100], decreased lifespan [36] among other consequences, and trade-offs for stress tolerance can extend to population effects such as sex-ratio bias [36] and transgenerational effects [101]. Both hypoxia tolerance and low pH tolerance are complex traits, with multiple physiological responses that must be coordinated, and the absence of phenotype covariance in them likely underscores the absence of a shared mechanistic or genetic basis for tolerance to these two factors.

## Supporting information

**S1 Table. Genetic divergence of cytochrome B among populations of *Tigriopus californicus*.** Pairwise distances were computed in MEGA [102] using the p-distance method. Sequences for AB, BB, SC, SD, and SS were downloaded from Genbank (accession numbers included). The sequence for FHL was determined from genomic sequencing. Accession number to be determined.
(DOCX)

**S2 Table. *Tigriopus californicus* collection locations, dates, and sample sizes.**
(DOCX)

**S3 Table. Preliminary observations of *Tigriopus californicus* tolerance of low oxygen, low pH, and high pH.** Low oxygen conditions were obtained by bubbling nitrogen gas into the water bath and monitoring swimming as described in the *Hypoxia and Low pH Assay* protocol. Low pH conditions were obtained by bubbling carbon dioxide gas into the water bath and monitoring swimming as described in the *Hypoxia and Low pH Assay* protocol. High pH conditions were obtained by adding concentrated sodium hydroxide solution to sea water and measuring swimming at the end of 24 h of exposure.
(DOCX)

**S4 Table. Observations of tide pool conditions containing *T. californicus* during August 2021.** Measurements from five tide pools at each of four latitudes along the California coastline were averaged by time and location. Standard deviations are given in parentheses. Measurements are divided by morning, afternoon, or evening. Locations are ordered from northernmost (BB) to southernmost (SD).
(DOCX)

**S5 Table. Averaged regression coefficients and importance of fixed effects from generalized-linear mixed effects models of hypoxia tolerance in *T. californicus*.** Significant *p*-values are written in bold. Regression coefficients are written with 95% confidence intervals. Importance is the weighted contribution of a predictor across all models compared. Importance values greater than 0.50 are noted with an asterisk ($^*$).
(DOCX)

**S6 Table. Averaged regression coefficients and importance of fixed effects from generalized-linear mixed effects models of low pH tolerance in *T. californicus*.** Significant *p*-values are written in bold, and regression coefficients are written with 95% confidence intervals. Importance is the weighted contribution of a predictor across all models compared. Importance values greater than 0.50 are noted with an asterisk (*).
(DOCX)

**S1 Fig. Diagram of experimental setup for hypoxia, low pH, and control assays.** The glove box was constructed of ¼" plexiglass, laser cut to measure 26 in (w) x 14 in (l) x 18 in (d) with a side window held with latch clamps. Seams and cracks were sealed with epoxy. Rubber kitchen gloves were held by hose clamps attached to 3 in to 4 in sewer drain adapters, tightly fitted and sealed with epoxy to the front of the glovebox. Laser cutter blueprints for the box are included on Dryad.
(TIF)

**S2 Fig. Diagram of DO and pH assay methods.** Black arrows indicate sampling time points.
(TIF)

**S3 Fig. Off-Gas rate in pH assays.** Mean pH (top) and DO (bottom) values during low pH assays are displayed with 1 SD bars. The pH and DO of the water bath prior to carbon dioxide injection is indicated in gray.
(TIF)

**S4 Fig. Bodega head state beach (BB) rockpool and ocean pH.** All pH measurements were taken with a handheld pH meter (Oakton Waterproof pH Testr® 30) in the winter of 2017–2018. Rockpool measurements were taken at sunrise and sunset, while ocean measurements were only taken at sunset. The blackline represents rockpool values whereas the blue line represents ocean values. The values of 6 rockpools were averaged. Bars are standard errors of the mean.
(TIF)

## Acknowledgments

We would like to thank A. Parker, M.E. Moore, K. Malinski, M. Alston, J. Lee and J.G. Kingsolver for their thoughtful feedback on the early drafts of this manuscript. We would also like to thank J. Rojas, M. Tojong, L. Scotto, H. Rendulich, and R. Pursley for their assistance in running assays and culturing copepods and J. Umbanhowar for his assistance with statistics. Finally, we are grateful to our reviewers whose feedback vastly improved the clarity and cohesion of the manuscript.

## Author Contributions

**Conceptualization:** Aimee Deconinck, Christopher S. Willett.

**Data curation:** Aimee Deconinck.

**Formal analysis:** Aimee Deconinck, Christopher S. Willett.

**Funding acquisition:** Christopher S. Willett.

**Investigation:** Aimee Deconinck.

**Methodology:** Aimee Deconinck, Christopher S. Willett.

**Project administration:** Christopher S. Willett.

**Resources:** Christopher S. Willett.

**Software:** Christopher S. Willett.

**Supervision:** Christopher S. Willett.

**Validation:** Christopher S. Willett.

**Visualization:** Aimee Deconinck.

**Writing – original draft:** Aimee Deconinck.

**Writing – review & editing:** Aimee Deconinck, Christopher S. Willett.

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
