## [Decision Letter · Decision Letter 0]

19 Jul 2022

PONE-D-22-16125Latitudinal cline in hypoxia tolerance does not result in correlated acid tolerance in Tigriopus californicusPLOS ONE

Dear Dr. Deconinck,

Thank you for submitting your manuscript to PLOS ONE. After careful consideration, we feel that it has merit but does not fully meet PLOS ONE’s publication criteria as it currently stands. Therefore, we invite you to submit a revised version of the manuscript that addresses the points raised during the review process. 

The manuscript received four reviews; all reviewers agreed that the manuscript is interesting and timely. Two reviewers recommended minor revisions (1 and 4), and two major revision (2 and 3). The reviews recommending minor revisions mostly focused on aspects that could strengthen the interpretation of the results and improve the quality of the discussion. In my view, these are excellent recommendations. The reviews recommending major revision reflect concerns on the quality/availability of the oxygen and pH data  and how they can bring into question the interpretation of the results. These are valid concerns.  If you have additional data that can put the concerns to rest, I encourage you to add them to the manuscript. Otherwise, the results may need to be reanalyzed and reinterpreted. At worse, the limitations of the experimental design and how they affect the results and their interpretation must be acknowledged. Reviewer 3 was also concerned about the validity of your correlations analysis.

Reviewer 3 provided some confidential remarks that may be a useful road map to rewriting the paper. I am comfortable sharing these remarks because I do not think that they affect the confidentiality of the review. Notice that some of the comments are similar to reviewer's 2.

" Their environmental data does not include night-time values (which is when you would expect the DO and pH minima to occur, thus linkages between the experimental design and field data are weak). The figures and results were not presented in a coherent way that allowed the reader to see what was significant (storyline lacking), the experimental design was really hard to tease out of the methods, and the discussion really didn’t discuss the results in the context of other pertinent literature. A lot of the supplemental material was important and could/should be condensed and added to the main MS so the reader can more easily follow along with all the pertinent information readily available. I also found it deceiving to discuss correlation between the low-O2 and low pH traits when they were not measured together (but separate single-stressor studies). I think that could be fixed by carefully fixing the text so that there is no confusion on this, but it also brings up the question of why the authors didn’t test these stressors together? Also, I think it’s quite problematic (maybe the biggest problem of the whole study) that the low pH trials they did were confounded by low O2. Even if the O2 levels were not at the experimental target (0.5 mg/L I think), 2 mg/L O2 during the low pH trials is still quite low. To me, this makes it very difficult to buy into the idea that there is ‘no evidence of correlation in tolerance for these two stressors’ Line 308.

They could rework this to discuss the single studies separately (low pH, low oxygen), and get rid of the correlation aspect, and discuss the environmental data more broadly at the end to say how multi-stressor studies are needed because low o2 and pH do actually co-occur. It’s likely going to involve a major rewrite" (end of quote)

If you do decide to submit a revised version of the manuscript, my intention is to call again upon the four reviewers and ask for a thumbs up or down on the fate of the manuscript.  Minor revisions after that are possible.

We look forward to receiving your revised manuscript.

Kind regards,

Hans G. Dam, Ph. D.

Academic Editor

PLOS ONE

Journal Requirements:

Reviewers' comments:

Reviewer's Responses to Questions

**Comments to the Author**

1. Is the manuscript technically sound, and do the data support the conclusions?

Reviewer #1: Partly

Reviewer #2: Yes

Reviewer #3: Partly

Reviewer #4: Yes

2. Has the statistical analysis been performed appropriately and rigorously? 

Reviewer #1: Yes

Reviewer #2: Yes

Reviewer #3: Yes

Reviewer #4: I Don't Know

3. Have the authors made all data underlying the findings in their manuscript fully available?

Reviewer #1: No

Reviewer #2: Yes

Reviewer #3: Yes

Reviewer #4: Yes

4. Is the manuscript presented in an intelligible fashion and written in standard English?

Reviewer #1: Yes

Reviewer #2: Yes

Reviewer #3: Yes

Reviewer #4: Yes

5. Review Comments to the Author

Reviewer #1: The manuscript covers a topic of great importance in the framework of impact of global change on marine organisms, and it would be very useful for those working in multiple-stressors, phenotypic plasticity, local adaptation, and the resilience of marine populations. However, I have some major concerns, regarding the, (i) theoretical background, (ii) experimental setup, and (iii) presentation of the results.

1. Page 3, Lines 40-44. These statements in the introduction are partially true, since the different behaviour of both gases, oxygen and CO2. While oxygen can increase during day time, CO2 cannot match with the same-interval due to differential diffusion and solubility of DO and CO2 can cause oxygen levels in seawater to come to equilibrium with prevailing atmospheric conditions more rapidly than CO2 (see Gobler & Baumann 2016). The resulting scenarios is a combination of scenarios, low pH/low O2, but also high O2 and low pH, due to the lag in the response of CO2.

2. Authors do not present simultaneously in an orderly and graphic way the natural fluctuations in both pH and oxygen in the sampled rockpools. The variability of O2 and pH are presented separately, when precisely the focus of this study is to discuss the context of change of both variables. The sampling frequency is quite irregular, and of very low frequency, which also does not allow a good characterization of the environment; however, it would be very helpful if they could re-analyze the data and present it in a way consistent with the objectives of the study.

3. For the experimental setup, it is not possible to follow the experimental conditions during the incubation. What were the temperature, pH and O2 evolution during each experiment... Did the pH level affected during experiment of hipoxia?, same for the low pH conditions?

4. More importantly, the study did not consider measurements at night hours, exactly the time interval when lowest pH and O2 occurs in these environments. During night conditions oxygen can drop to almost zero in rocky tidal pools, pH lowest than 7.3 and and CO2 can increase up to > 3000 uatm (See Kwiatkowski et a. 2016). Since extreme levels in each rockypool along this cline play a disproportionately large role in shaping the physiology, ecology and evolution of these populations, I can see some part of this history is missed, and it is difficult for me to analyze any correlation with latitudinal patterns if we lack of extreme environmetnal conditions among studied populations.

Reviewer #2: General Comments:

This manuscript looks at effects of body size, latitude (collection location), and sex on hypoxia tolerance and low pH tolerance in the copepod Tigriopus californicus. This paper examined how copepod cultures from six locations ranging from San Diego, CA to Friday Harbor Labs in Washington survive when exposed to acute exposures of low oxygen concentration and low pH. The authors hypothesized 1) that both stressors co-occuring would yield population-specific tolerance to either stressor, 2) that hypoxia and pH tolerance would inversely correlate with latitude, and 3) one of the evaluated effects would predict tolerance. The authors found that while latitude and length are predictors of performance with respect to hypoxia tolerance, only sex is a predictor for pH tolerance. Patterns of hypoxia tolerance with respect to latitude are converse to the authors expectations and may reflect duration of lab rearing rather availability of oxygen.

This is a very interesting study. I have few comments on the nature of this work. It is well designed, executed and analyzed. My only criticism would be the interpretation. There is nothing wrong with how the data is interpreted as it is, but it could benefit from more explanation or a deeper dive into why the authors believe they observed the patterns they did, particularly when considering the hypoxia tolerance pattern with latitude. I recommend the authors consider a few more ideas as to why hypoxia tolerance increases with latitude rather than decreases. Is there an evolutionary pattern that may be at play? Is the tolerance strictly plastic?

Specific comments:

Lines 49-50, 308, 326-327: Fix references to match citation style.

Lines 204-206: “Importance” comes up a lot in this manuscript. How does one evaluate what the critical level of importance is? In other words, how important is “important”? Does a value >0.5 suggest that it is somehow not random?

Table 2 – A figure depicting the temperature, DO, and pH changes across a day for each location would be easier to interpret. If possible, the authors should consider making one.

Lines 249-250: “…although the latter was opposed by the effect of latitude.” What does it mean for one effect to oppose another? Is this meant to represent an antagonistic relationship?

Lines 310-315: So, hypoxia tolerance is a physical attribute and pH tolerance is a physiological one? This seems interesting. I suggest the authors discuss how both tolerance systems may involve physical and/or physiological mechanisms. For instance, the upregulation of oxidative stress proteins does involve certain reactive oxygen species that can be accumulated from hypoxia, not just low pH. The authors should expand on other stress tolerance mechanisms that are different for each of the stressors discussed in the manuscript.

Lines 324-335: This is very interesting and deserves more discussion. Please include more detail as to what the “temporal” correlations are. Are they across years? Or across the day?

Reviewer #3: Latitudinal cline in hypoxia tolerance does not result in correlated acid tolerance in Tigriopus californicus

This paper aims to determine if there is a correlation between low oxygen and low pH tolerance in populations of copepods from a broad latitudinal range. The topic is of great importance to the field and a better understanding of how the temporal variability of these two stressors impacts marine animals is needed. Including latitude and population-level responses is especially helpful, and difficult to do, making this contribution important. However, there are some issues with the current draft that should be addressed before publication. I hope my comments below are clear and helpful.

• The environmental data presented did not include continuous or night-time values. This is when you would expect the DO and pH minima to occur, thus linkages between the experimental design and field data are weak, and are the assumptions based on them. There isn’t much information about conditions as they relate to the latitudinal cline either.

• If I understood correctly, the low pH trials are confounded by co-occurring low O2. Even if the O2 levels were not at the experimental target (0.5 mg/L I think), 2 mg/L O2 during the low pH trials is still quite low (and hypoxic as this paper defines it!). To me, this makes it very difficult to say that there is ‘no evidence of correlation in tolerance for these two stressors’ Line 308. This should be discussed and addressed, and the correlation, and its interpretation, between the low O2 and pH trials should be reconsidered.

• Additionally, I found it misleading to discuss correlation between the low-O2 and low pH traits when they were not measured together (but separate single-stressor studies). I think that could be fixed by carefully fixing the text so that there is no confusion that a multiple stressor study was not performed, but single stressor trials. It also brings up the question of why the authors didn’t test these stressors together? There are many examples of multiple stressors (together) having very different, and important physiological impacts compared to single stressors.

• It would be helpful if figures were reworked to help the reader to see the significant impact hypoxia and low pH (separately) have across populations/sites. A lot of the supplemental material was important and could/should be condensed and added to the main MS so the reader can more easily follow along with all the pertinent information readily available (e.g., a map that shows where the sites are and how they are environmentally distinct).

• It might be helpful to discuss the results from the low pH trial and the low oxygen trial separately and put less emphasis on the correlation analysis.

Minor points:

Line 26 – both stressors when tested together? Will be helpful to differentiate single vs combined stress responses throughout the text

Citation style format might be correct for journal, but seems to vary between numbers (13) and names (Edmands, 2018) throughout the text.

Line 57 add references for this. Also, specify that these multiple studies are (or are not) linking temperature, hypoxia and acid tolerance together as an adaptation. It’s confusing because I think you are trying to say that there are not studies on this topic (which I agree with and is the point you should emphasize -because your study is one of the first to test this). The first sentence of this paragraph doesn’t really say what I think you are trying to say here.

‘Low pH’ is a better word to use than ‘acid’ – I would change this and be consistent throughout. As is, there is a mixture of terminology used.

Also, define ‘hypoxia’ is border-line jargon without an exact definition, make sure this is defined once and used in that context throughout.

Line 63- is Pc Pcrit?

Line 88 – this is implies that both stressors are tested together, not singularly, this should be clarified

Line 92-98- It would be helpful to add a map of all the locations, how far away from each other are these sites? The field measurement section here also could benefit from a figure showing the diurnal flux at the different sites, is it approx. the same magnitude along the gradient?

The field data should be shown in the main text, maybe on the map, to highlight differences between sites

Line 109- where these copepod cultures all being kept at the Bodega Marine Lab?

Line 114- up until now I thought the study was a multiple stressor study, not two single stressor studies. It would be helpful to ensure the experimental design is clear right away (in the intro). Also, was a multiple stressor (i.e., combined stressor) treatment attempted? If not, why? There can be different responses (synergistic, additive, antagonistic) when the stressors occur in combination, so some discussion on this missing element in the study would be beneficial.

Table 1 could be converted into a map, and sent to the supplemental materials

Line 117- where did experiments take place?

Line 148- experimental design. It would be helpful if the authors briefly mentioned the experimental design earlier in the intro and methods. Also, please specify earlier what the target DO/pH levels were.

How often were DO measurements were made during the trials (same for pH?), and what the variability in both pH and DO during these trials?

How was the oxygen kept stable?

It is intriguing to me that the pH went up during these trials, what was the starting pH? Why would this be?

Line 152- can the authors elaborate on this experiment and how/why they chose the pH and DO values for the current study? It would be very helpful to be more specific about the exploratory experiments -this is interesting. – i.e., ‘75% of tested copepods were able to swim after xx hour exposure to oxygen concentrations of xx mg/L, therefore we chose to xx’

The pH values seem very high to me. Is this normal?

Line 157- Please add the accuracy/precision of this DO probe and describe how it waw calibrated. Was this done with a 0-point calibration solution?

Line 156- hypoxia was maintained for 20 hours- does this include the ~2 hours of ramping (when the nitrogen was turned on?)

Line 163- again, more details would be very helpful about the previous experiments

Line 161 – so the animals were exposed to 24 of low pH and 20 hours of low DO?

Line 166- was this the same for the DO trial?

Line 182- how many copepods were not recovered? Why was this? Please put a number as to how many were lost. This is a concern if they were escaping from their cages.

Figure S5 and S6- It would be useful to add time to these plots, to help identify what time of day you are referring to. Sunset and sunrise? What about noon? S6 is missing DO units. Also, would be good to have standard deviation for both of these to make comparisons easier.

Line 219- again, it would be nice to have a map with this information in it, showing the variation in pH and DO in tidepools along the latitudinal gradient. It would be much easier to understand if Table 2, and Fig S5 and S6 were combined into one figure, in the main MS.

Line 231- what about night-time values? There are a lot of studies and data showing the pH and oxygen minima occur at night due to respiration. This is missing, and an important aspect to making claims such as this about DO, pH and latitudinal trends. Authors should take care in making interpretations from their measurements, or add additional data from other sources to help.

Line 236 – Please clarify what ‘recovery’ is referring to. It’s still unclear what the pH trial duration was, and if there was a recovery phase like that in the oxygen trials.

Line 248 – is hypoxia survival significantly different between populations/latitudes? Please add details to text here, as well as Figure 2.

Line 248- Were females more tolerant than males in each population, under both low pH and oxygen? More details, including statistics, would be useful here. I suggest taking the pertinent info from the tables and adding to the text, and then the tables can be added to the SM.

Figure 3 caption says sex was important but not significant. What does that mean exactly? Additionally, why arrange panels based on collection and not latitude like Figure 2? It would be great to be consistent throughout. This also draws attention away from the latitudinal differences in hypoxia survival.

Line 277-280- information for the methods

Line 287: It would be useful to link the findings of this study to the extensive literature on the topic of diurnal flux in temp, pH and DO (Gobler and Baumann has some good references in general, but I’m sure others exist for tide pools)

Line 323- Thermal tolerance is mentioned here, can the authors explain what this is referring to, what study, add details to this, etc.?

When referring to hypoxia or acid tolerance it would be better to just say the trait that is being assessed (i.e., survival) and eliminate any uncertainty as to what is being discussed in terms of ‘tolerance. ‘

Reviewer #4: Deconinck et al test for potential correlation in phenotypic response to two diurnally correlated stressors in six geographically distinct populations of Tigriopus californicus and the effect of sex, body size, and collection time on stress tolerance using a combination of laboratory based experiments and statistical modeling. The authors’ use of self-designed custom glove boxes to conduct their experiment is commendable. The paper is well suited for the audience of PLOS One. My comments largely focus on the experimental sections of this study, as computational modeling is not my area of expertise.

Abstract

The first sentence is a bit awkward – consider rephrasing.

Introduction

45-46 – This sentence needs a reference

55-57 – This sentence needs a reference

71 – You should cite Ligouri 2022 for an important counter-example to the other studies suggesting that low pH is stressful in copepods.

Liguori, Alyssa. "Multigenerational Life-History Responses to pH in Distinct Populations of the Copepod Tigriopus californicus." The Biological bulletin 242.2 (2022): 97-117.

Although Table 2 does provide detailed records of field DO and pH conditions, it would be helpful to state the natural pH and DO range experienced by T. californicus in the first two paragraphs (Line 32 - 54) itself, especially for readers who might not be familiar with the study system.

Similarly, readers would find it easier to interpret sampling details (Table 1) and information from Table 2 if it was conveyed in a graphical manner (with locations plotted on a geographical map of the region), while retaining both the tables in the supplementary information section.

The introduction section could be strengthened by including examples of other species that show correlated phenotypic response to co-occurring stressors.

Methods:

Were copepods exposed to both stressors simultaneously or separately?

How was the exposure time chosen for hypoxia and low pH assays?

A picture of the custom glove-box chamber or its pictorial representation would be helpful to orient your readers to your unique experimental set-up.

Results and Discussion:

The authors should consider justifying their choice of using survival as the sole fitness measures. The results and discussion sections also do not mention potential trade off mechanisms that could affect survival and other traits in an organism. For example, is it possible that the survivor animals from the stress trials had lower number of offsprings than the control animals? Relying on only one fitness measure also lends very little credit to line 367 as the authors are essentially using only one phenotypic measure to strike down the possibility of genetic covariance in their study system. I strongly recommend including more phenotypic measurements to support the claim in line 367.

238 – pH exposures are substantially lower than natural pH values for Tigriopus pools. Somewhere you need to better justify the use of such low pH values (besides the last paragraph of the discussion, which states that other studies have also done this…)

6. PLOS authors have the option to publish the peer review history of their article (what does this mean?). If published, this will include your full peer review and any attached files.

Reviewer #1: No

Reviewer #2: **Yes: **James A. deMayo

Reviewer #3: No

Reviewer #4: No

---

## [Author Response · Author response to Decision Letter 0]

15 Sep 2022

Detailed responses to the reviewers comments are included in the "Response to reviewers" document.

---

## [Decision Letter · Decision Letter 1]

11 Oct 2022

Hypoxia tolerance, but not low pH tolerance, is associated with a latitudinal cline across populations of Tigriopus californicus

PONE-D-22-16125R1

Dear Dr. Deconinck,

We’re pleased to inform you that your manuscript has been judged scientifically suitable for publication and will be formally accepted for publication once it meets all outstanding technical requirements.

Kind regards,

Erik V. Thuesen, Ph.D.

Academic Editor

PLOS ONE

Additional Editor Comments (optional):

Reviewers' comments:

Reviewer's Responses to Questions

**Comments to the Author**

1. If the authors have adequately addressed your comments raised in a previous round of review and you feel that this manuscript is now acceptable for publication, you may indicate that here to bypass the “Comments to the Author” section, enter your conflict of interest statement in the “Confidential to Editor” section, and submit your "Accept" recommendation.

Reviewer #2: All comments have been addressed

2. Is the manuscript technically sound, and do the data support the conclusions?

Reviewer #2: Yes

3. Has the statistical analysis been performed appropriately and rigorously? 

Reviewer #2: Yes

4. Have the authors made all data underlying the findings in their manuscript fully available?

Reviewer #2: Yes

5. Is the manuscript presented in an intelligible fashion and written in standard English?

Reviewer #2: Yes

6. Review Comments to the Author

Reviewer #2: Thank you for addressing the comments. This is an interesting experiment that I hope will lead to future studies examining trends in hypoxia and pH tolerance in tide-pool copepods.

7. PLOS authors have the option to publish the peer review history of their article (what does this mean?). If published, this will include your full peer review and any attached files.

Reviewer #2: **Yes: **James deMayo

---

## [Editor Report · Acceptance letter]

20 Oct 2022

PONE-D-22-16125R1 

Hypoxia tolerance, but not low pH tolerance, is associated with a latitudinal cline across populations of Tigriopus californicus 

Dear Dr. Deconinck:

I'm pleased to inform you that your manuscript has been deemed suitable for publication in PLOS ONE. Congratulations! Your manuscript is now with our production department. 

Kind regards, 

on behalf of

Dr. Erik V. Thuesen 

Academic Editor

PLOS ONE